# Using an objective computer task (QbTest) to aid the identification of attention deficit hyperactivity disorder (ADHD) in the Children and Young People Secure Estate (CYPSE): a feasibility randomised controlled trial

Prathiba Chitsabesan ![ORCID],[1] Charlotte Lucy Hall ![ORCID],[2] Lesley-Anne Carter,[3] Mindy Reeves,[4] Vaseem Mohammed,[4] Bryony Beresford,[5] Susan Young,[6,7] Abdullah Kraam,[8] Sally Trowse,[9] Lloyd Wilkinson-Cunningham,[10] Charlotte Lennox ![ORCID] [11]

For numbered affiliations see end of article.

**Correspondence to**
Dr Prathiba Chitsabesan;
pchitsabesan@nhs.net

## ABSTRACT

**Objectives** QbTest has been shown to improve time to decision/diagnosis for young people with attention deficit hyperactivity disorder (ADHD). The aim was to assess the feasibility of QbTest for young people in prison.

**Design** Single-centre feasibility randomised controlled trial (RCT), with 1:1 allocation. Concealed random allocation using an online pseudorandom list with random permuted blocks of varying sizes.

**Setting** One Young Offenders Institution in England.

**Participants** 355 young people aged 15–18 years displaying possible symptoms of ADHD were assessed for eligibility, 69 were eligible to take part and 60 were randomised.

**Intervention** QbTest—a computer task measuring attention, activity and impulsivity.

**Main outcome measures** Eligibility, recruitment and retention rates and acceptability of randomisation and trial participation.

**Results** Of the 355 young people assessed for eligibility, 69 were eligible and 60 were randomised (n=30 QbTest plus usual care; n=30 usual care alone). The study achieved the specified recruitment target. Trial participation and randomisation were deemed acceptable by the majority of participants. 78% of young people were followed up at 3 months, but only 32% at 6 months, although this was also affected by COVID-19 restrictions. Secondary outcomes were mixed. Participants including clinical staff were mostly supportive of the study and QbTest; however, some young people found QbTest hard and there were issues with implementation of the ADHD care pathway. There were no serious adverse events secondary to the study or intervention and no one was withdrawn from the study due to an adverse event.

**Conclusions** With adaptations, a fully powered RCT may be achievable to evaluate the effectiveness of QbTest in the assessment of ADHD in the Children and Young People Secure Estate, with time to decision (days) as the primary outcome measure. However, further programme

## STRENGTHS AND LIMITATIONS OF THIS STUDY

⇒ The study has contributed to developing the evidence base in understanding barriers and facilitators to implementing QbTest within a secure estate setting.

⇒ The study produced important feasibility information on eligibility, recruitment, attrition rates and collection of primary outcomes.

⇒ The sample was obtained from only one site within the secure estate and includes boys only.

⇒ While this was a pragmatic trial to assess feasibility, there were challenges identified in the usual care pathway and the number of diagnostic decisions within the follow-up period was limited.

⇒ COVID-19 restrictions had an impact on delivery of usual care, access to the prison and collection of some of the outcome measures.

developmental work is required to address some of the challenges highlighted prior to a larger trial.

**Trial registration number** ISRCTN17402196.

## BACKGROUND

Attention deficit hyperactivity disorder (ADHD) is a common mental health disorder; significantly greater in prevalence in the Children and Young People Secure Estate (CYPSE) up to 30.1%[1]; than the general population of young people 5%.[2] However, a recent report raised significant concerns about the number of young people in the CYPSE with undetected neurodisability, including ADHD, due to a lack of appropriate screening and assessment processes and training of staff.[3]

A clinical assessment of ADHD requires integration of information including observation and reports from parents, teachers and young people. This can be difficult to obtain for young people in the CYPSE. This approach is also reliant on subjective measures that can lead to a lack of reliability and consistency in the diagnosis of ADHD[4] or delays when there is a difference of opinion between informants.[5] All these factors can lead to delays in receiving a diagnosis and accessing evidence-based treatment. Early diagnosis and timely interventions reduce the risk of adverse long-term outcomes that are associated with ADHD such as antisocial behaviour, poor academic performance and social functioning.[6] The social and economic burden of untreated ADHD on society is significant[7–9] and is estimated to be around £100 000 per case,[10] although this does not include any criminal justice costs. There are effective treatments for ADHD including medication that can improve outcomes for young people.[11 12] One way of supporting the assessment for ADHD is the use of objective measures.

One objective measure is the continuous performance test (CPT). There are several variants of this but all measure vigilance and sustained attention. While CPT can demonstrate good sensitivity to ADHD and correlates well with symptoms,[13] other studies have shown significant overlap in the performance of children with ADHD and typically developing children[14] and variability in intellectual ability may confound the interpretation of CPT performance in ADHD.[15]

However, recent evidence suggests that combining a CPT with an objective measure of motor activity may add value in the clinical assessment of ADHD.[16 17]

QbTest (Qbtech) combines a computerised CPT with an infrared camera to detect motor activity during the test and provides an objective standardised measurement of attention, impulsivity and activity, QbTest is highly correlated with blinded observer ratings of ADHD symptoms in placebo-controlled trials[18] and can help differentiate ADHD from other conditions.[19]

In studies designed to assess 'stand-alone' diagnostic accuracy, QbTest has only moderate sensitivity and low specificity to ADHD.[20] Additionally, there are concerns about QbTest identifying ADHD in clinical samples.[21] However, these studies used QbTest independently of other clinical information. QbTest is not designed to act as a 'stand-alone' tool; the US Food and Drug Administration approved QbTest as a decision-aid tool to augment, but not replace, standard clinical assessment of ADHD by a trained practitioner.[11]

A recently published single-blind randomised control trial[22] demonstrated positive results for the role of Qbtest in supporting ADHD diagnosis alongside a routine ADHD assessment in a clinic referred sample. This assessment approach has been shown to be acceptable to both families and clinicians.[23] At 6 months, clinicians with access to the QbTest report were more likely to reach a diagnostic decision about ADHD, felt more confident in their decision and took less consultation minutes to reach the decision compared with those who had not seen a QbTest report. There was no difference in diagnostic accuracy. The authors concluded QbTest may increase the efficiency of ADHD assessment pathway allowing greater patient throughput with clinicians reaching diagnostic decisions faster without compromising diagnostic accuracy.[22] One suggested approach to using CPTs is in a stepwise approach to support the diagnostic process at a later stage when there is clinical uncertainty.[24] While this may be a feasible approach in community settings, it may delay diagnostic decision-making and young people only remain in the secure estate for relatively short periods of time.

## Aims and objectives

To assess the feasibility and acceptability of conducting a pragmatic randomised controlled trial (RCT) of QbTest in the assessment of ADHD in the CYPSE.

## METHODS

### Trial design

We conducted a single-centre parallel two-group feasibility RCT with 1:1 individual participant allocation to QbTest plus usual care or usual care alone. Recruitment started on 26 March 2019 and ended on 27 February 2020. Six-month follow-up continued until October 2020. The primary aim of the study was to assess the feasibility and acceptability of conducting a pragmatic RCT of QbTest in the assessment of ADHD in the CYPSE. We were interested to evaluate whether outcomes measures could be reliably collected at baseline and follow-up and to help identify primary and secondary measures for a larger RCT.

### Participants

The study took place in one Young Offenders Institution (YOI) accommodating boys aged 15–18 years in England. Participants were eligible if they answered a 'yes' to any of the ADHD symptoms on the Comprehensive Health Assessment Tool (CHAT) assessment (The CHAT is a health needs assessment used in the CYPSE at the point of admission. It assesses first night risk, eg, self-harm, followed by a comprehensive assessment of physical and mental health, substance use and neurodisability over the first 10 days of admission. The ADHD section on the CHAT contains symptom questions around concentration, restlessness/fidgeting, interrupting and difficult waiting turn.). Participants were excluded if: they were on remand, did not speak English, had a previous or current diagnosis of ADHD, deemed a risk to either the researcher or staff, unable to provide informed consent (over 16 years) or parent/legal guardian consent was not received (under 16). Consent was obtained by the research assistant. The flow of participants is shown in figure 1. A sample size of 60 (30 to each group) were large enough to test the feasibility of the research procedures.[25]

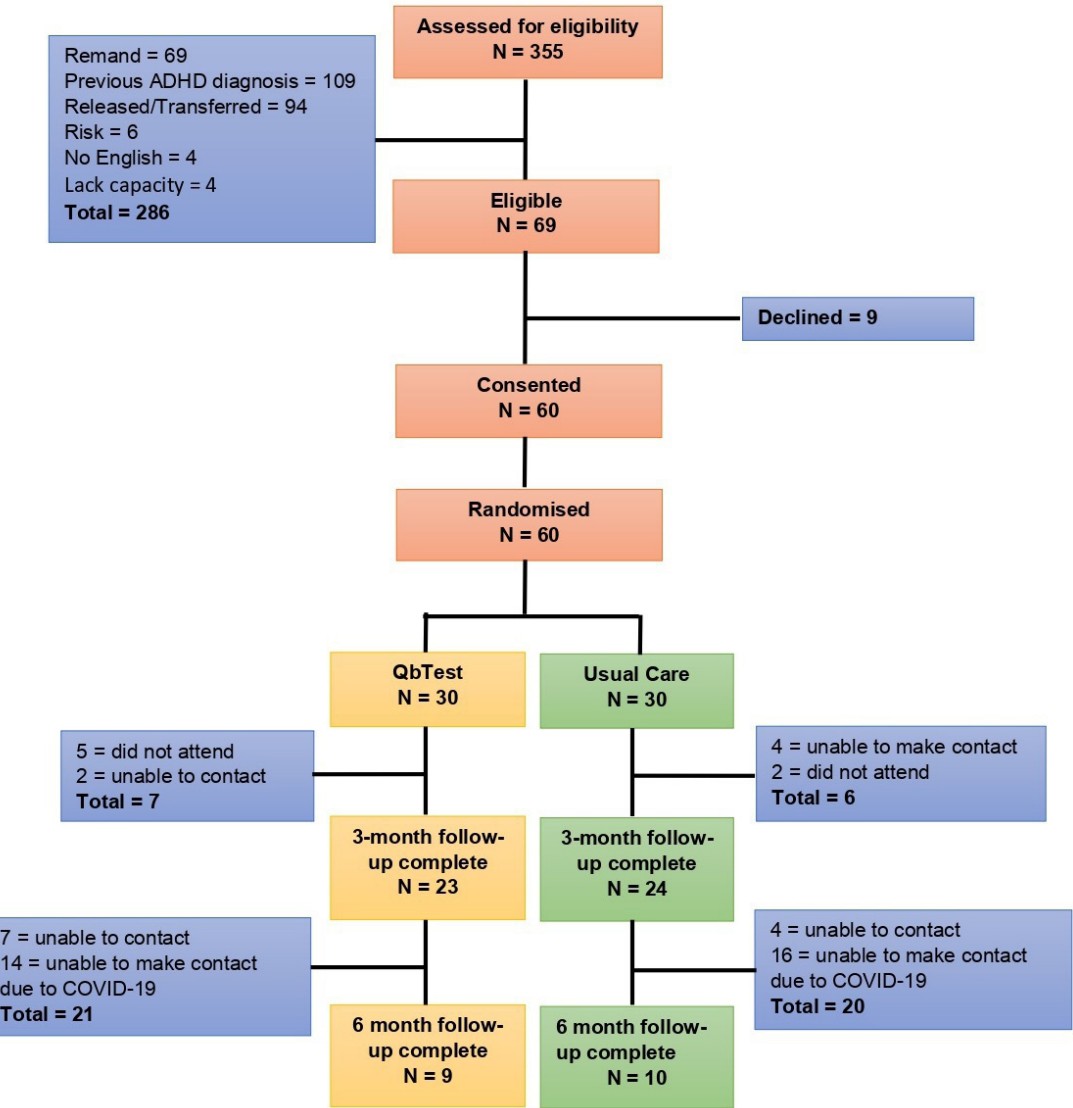

**Figure 1** QbTest Consolidated Standards of Reporting Trials chart. ADHD, attention deficit hyperactivity disorder.

## Usual care

Within the YOI, the expected usual care was as follows: young people identified with potential ADHD are offered an initial appointment by the neurodevelopmental lead, and initially assessed to decide whether the young person potentially has ADHD. This appointment was expected to be offered within a month of randomisation. If there were potential ADHD symptoms, then an assistant mental health practitioner would commence the assessment by completing questionnaires, collecting a developmental history and observing the young person. The time frame for this varies, but would aim to be completed within 2 months of randomisation. Once the history, observation and informant information are collected, another appointment would be made for the young person with the neurodevelopmental lead to review the information and either diagnose with ADHD or exclude diagnosis. If a diagnosis were made, then medication would be trialled, and the young person monitored for any improvements in presentation or side effects from medication.

## Intervention: usual care plus QbTest

Prior to the triage appointment by the neurodevelopmental lead, young people in the QbTest group were booked into the QbTest clinic. QbTest (Qbtech) is a computer task that measures three core aspects of ADHD: attention, impulsivity and motor activity. Performance on the task provides information (via an immediate report) on the three symptom domains of ADHD and a 'summary score' based on deviation from a normative data set (for age and gender). The healthcare team conducted the QbTest and used the information from the QbTest report in conjunction with the clinical information to inform their decision whether the young person had ADHD or not.

## Randomisation

Randomisation was completed by the research assistant and achieved by means of concealed random allocation using an online pseudorandom list hosted by Sealedenvelope.com with random permuted blocks of varying sizes.

## Blinding

Given the nature of QbTest, it was not possible to blind participants or those involved in delivering the intervention. Participants and research team were blind to allocation at the time of the baseline measures being completed, which occurred before randomisation. It was not possible to maintain the field researcher being blind to trial arm allocation because the researcher and staff delivering the intervention were both working within the confines of the YOI. Only the statistician (L-AC) who was assessing the outcomes was blind to trial arm allocation.

## Primary outcomes

The following outcome measures and data were collected:

▶ Eligibility rate recorded as the number of eligible young people against the total number of young people assessed for eligibility (ie, identified with ADHD needs on the CHAT).

▶ Recruitment rate recorded as the number of young people who consent to participate against the number of eligible young people.

▶ Acceptability of randomisation recorded as the number of young people randomised.

▶ Acceptability of trial participation recorded as the number of eligible young people who dropped out after receiving allocation.

▶ Retention rate recorded as the number of young people who were randomised that remain in the study until the end of follow-up at 6 months (target of 50% at 6 months in both trial arms).

## Secondary outcomes

The secondary outcome measures selected included measures of ADHD symptoms, quality of life and service use. Other key data collected such as the number of appointments and days until diagnosis and number of behavioural episodes were also key outcome data that would be relevant to a larger study in measuring the impact of the intervention arm (QbTest) and identifying differences between the two groups (usual care vs usual care and QbTest group). The following outcome measures and data were collected at baseline, 3 and 6 months, unless otherwise stated:

Completed by the young person with support from the research assistant.

▶ Behaviour measured using the Strengths and Difficulties Questionnaire (SDQ).[26] The SDQ is a well validated and commonly used brief screening questionnaire of children's emotions and behaviours through five scales (eg, emotional, conduct problems, hyperactivity/inattention, peer relationship problems and prosocial) scored 0–10. Scores are categorised from 'close to average' (80% of population) to 'very high' (5% of population).

▶ ADHD symptoms were measured using the Brief Barkley Adult ADHD Rating Scale (B-BAARS).[27] The B-BAARS has been developed as a brief assessment to screen for ADHD in adults in the criminal justice

system and has been shown to accurately predict the likelihood of receiving a clinical diagnosis using DSM-V (Diagnostic Statistical Manual of Mental Disorders, 5th Edition) criteria.

▶ Health-related quality of life was measured using the Child Health Utility Instrument (CHU-9D)[28] (Only reported as level of missing data.). The CHU-9D is a quality of life measure containing nine main dimensions (worried, sad, pain, tired, annoyed, schoolwork/homework, sleep, daily routine and activities), each with five increasing levels of severity/impairment.

▶ Contact with services was measured using the modified Client Service Receipt Inventory.[29] It allows resource use patterns to be described and support costs to be estimated using an appropriate unit cost.

▶ At 3 months, feedback from young people receiving the QbTest using the QbTest opinion questionnaire.

Completed by the clinical team.

▶ The Children's Global Assessment Scale (C-GAS)[30] is a rating of functioning for young people 6–17 years of age. The young person is given a single score between 1 and 100 based on a clinician's assessment of the young person's psychological and social functioning. The score puts them in 1 of 10 categories that range from 'extremely impaired' (1–10) to 'doing very well' (91–100).

▶ The number of appointments until confirmed ADHD diagnosis/exclusion of ADHD and the number of days until a decision was reached was recorded on a pro-forma completed by clinicians after each appointment with the young person.

Completed by the teacher.

▶ The Swanson, Nolan and Pelham Rating Scale (SNAP-IV) is a widely used scale that measures the core symptoms of ADHD.[31] Teachers are asked to rate 26 items on a 4-point scale.

Completed by the research assistant from health and prison records.

▶ A number of recorded behavioural incidents were collected using prison information.

▶ Numbers of young people agreeing to sit the QbTest and the number who completed the QbTest was recorded.

Qualitative interviews with young people and staff.

▶ At 3 months, participants were interviewed using a semistructured interview schedule about the acceptability of randomisation, acceptability of outcome measures and for those in the QbTest group acceptability of the QbTest. Purposive sampling was used to select participants for interview including consideration of age, completion of the QbTest and scores on the QbTest opinion questionnaire.

▶ At the end of the study, health professionals and CYPSE staff were interviewed on the acceptability and feasibility of administering and implementing QbTest within usual assessment practices as well as facilitators and barriers to using the QbTest and reasons for non-completion.

See study protocol[32] for details of how data collection was conducted.

## Data analysis

As a feasibility study, data analysis was mainly descriptive.[33] All measures were summarised by group across follow-up time with mean and SD for normally distributed data, median for skewed variables and frequency (percentage) for categorical data. All statistical analysis were conducted using Stata V.15.

Interviews were conducted and recorded by the researcher using an encrypted Dictaphone. All qualitative data were analysed using thematic analysis[34] with the aid of NVivo. Data were coded inductively into themes, creating a detailed coding scheme, allowing for the investigation of emergent patterns between individual codes and boarder emergent themes.

## Patient and public involvement (PPI)

PPI was embedded in many aspects of the study including; the development of the funding application and shaping the study objectives and outcome measures. The burden of the intervention for young people had been previously assessed through a local community audit. Within the project, we supported the development of a PPI group within the YOI and they met on a quarterly basis,

facilitated by the research assistant. This group helped develop our patient information sheet and advised on recruitment and data collection including the need for the researcher to be flexible in their approach, for example, seeing young people on the unit rather than the young people attending healthcare. COVID-19 significantly impacted our ability to continue this group and as a result, our PPI group were not involved in the analysis or dissemination stages of the project as we had planned. However, our PPI representatives (two parents of young people with ADHD) have helped us interpret the qualitative data and write lay summaries and infographics for staff and young people. This will be disseminated to staff through a webinar and written summary and to the young people through the healthcare team.

## RESULTS

A total of 60 young people were randomised 30 to QbTest plus usual care (intervention) and 30 to usual care (control). Table 1 shows the demographics of the 60 boys randomised to the study. Of the 60 young people a subsample of 11 young people (6 QbTest vs 5 usual care) were interviewed. The research assistant and five members of staff were also interviewed, including all those who were delivering QbTest and staff involved in supporting the implementation.

## Primary outcomes

### Eligibility rate

Participant flow (see figure 1) showed that over 12 months we assessed 355 young people for eligibility. The numbers and reasons for ineligibility were: 69 (19%) on remand, 109 (30%) had a previous diagnosis of ADHD, 94 (26%) were released or transferred before they could be approached, 6 (2%) were deemed too high risk, 4 (1%) were not fluent in English as a language and 4 (1%) were deemed to lack capacity. This gave us an eligibility rate of 19.4% (69/355).

### Recruitment rate

Of the 69 young people who were eligible, 60 (87%) consented to participate in the study. For the nine participants who did not consent, their reason was either they did not feel they had ADHD or that they were leaving prison soon. Our initial estimates were to achieve an average recruitment rate of five randomisations a month (60 young people over 12 months) which was successfully achieved.

### Acceptability of randomisation and trial participation

Of the 60 young people who consented to take part all agreed to be randomised and no young people refused to continue in the study after randomisation (see figure 1).

### Retention rate

Of the 60 young people, 47 were followed up at 3 months, a retention rate of 78% which exceeded our target. Our aim was to successfully follow-up at least 50% at 6 months. However, we were only able to

| Table 1 | Baseline demographic information | | | |
|---|---|---|---|---|
| | QbTest (n=30) | | Usual care (n=30) | |
| | n | % | n | % |
| Age | | | | |
| 16 | 6 | 20.0 | 3 | 10 |
| 17 | 8 | 26.7 | 11 | 36.7 |
| 18 | 15 | 50.0 | 16 | 53.3 |
| Missing | 1 | 3.3 | 0 | 0 |
| Ethnicity | | | | |
| White | 23 | 76.7 | 24 | 80.0 |
| Other | 6 | 20.0 | 6 | 20.0 |
| Missing | 1 | 3.3 | 0 | 0 |
| Accommodation | | | | |
| House or flat | 20 | 66.7 | 25 | 83.3 |
| Other | 9 | 30.0 | 5 | 16.7 |
| Missing | 1 | 3.3 | 0 | 0 |
| Lived with | | | | |
| Parent(s) | 15 | 50.0 | 21 | 70.0 |
| Other | 13 | 43.3 | 9 | 30.0 |
| Missing | 2 | 6.7 | 0 | 0 |
| Education | | | | |
| Mainstream | 6 | 20.0 | 6 | 20.0 |
| Pupil referral unit | 3 | 10.0 | 5 | 16.7 |
| None | 20 | 66.7 | 17 | 56.7 |
| Other | 0 | 0 | 2 | 6.7 |
| Missing | 1 | 3.3 | 0 | 0 |

follow-up 19 of the 60 young people; a retention rate of 32%. Half of the 6-month follow-ups were due after March 2020 and therefore were significantly impacted by COVID-19 as we were not able to access the YOI from March 2020 for 8 months.

There were no serious adverse events secondary to the study or intervention and no one was withdrawn from the study due to an adverse event.

### Secondary outcomes

Of the 30 young people randomised to receive QbTest, all agreed to sit the QbTest. At the end of the recruitment window, four young people could not undertake their QbTest due to the COVID-19 lockdown in the prison. For the remaining 26 young people, a total of 6 (23%) did not complete their QbTest: for 3 young people (12%) the appointment was not attended due to prison resources (staffing to accompany the young person), 2 (8%) young people were released/transferred before the QbTest could take place and one young person (4%) was unable to sit the QbTest due to concerns from the clinical team regarding increased risk of self-harm at the time. The median number of days between randomisation and QbTest was 42 (IQR=26–93; min=1; max=195).

Table 2 shows the descriptive statistics for the questionnaires collected at baseline, 3 and 6 months for the two trial arms from the young people.

The amount of missing data for all outcomes are shown in table 2. Of those contactable for the 3-month follow-up, 64%–66% completed the SDQ, B-BAARS and CHU-9D. Completion rates of the 6-month measures were higher; however, only 19 participants were contactable. The C-GAS and SNAP were poorly completed across all time points. As the study aimed to assess the feasibility of collecting outcome measures, we have not made comparisons of the data between intervention arms and over time points.

Of the 60 young people, a total of 32 (53%) received at 1 appointment and 3 (5%) had two appointments. Therefore, a total of 25 young people (42%) did not receive any appointment within the 6-month follow-up window. Some young people did have appointments, which fell outside of the 6-month window. Appointments had been offered to some young people but were not attended (n=53).

The low number of appointments offered and attended within the 6-month follow-up period impacted on the number of decisions being made. We recorded evidence of 14 decisions across the 60 participants, 8 in the QbTest group and 6 in the usual care group (27% event rate vs 20%). All these decisions were exclusions of ADHD; there were no diagnoses of ADHD for any young people.

Data from the qualitative interviews suggested that most of the young people had some understanding of the randomisation process and were able to explain what had happened. Most of the young people said that they were happy or were not upset about being allocated to either group. There were, however,

two young people who expressed stronger feelings about the group to which they were allocated. One was angry about being allocated to the QbTest group because they did not like the test as it took too long, and the other was allocated to usual care but wanted to receive QbTest as they thought they had ADHD. The young people were all happy with being involved in the research and one young person said that they would be happy to recommend the study to others.

Of the 20 young people who had received QbTest, 10 completed the QbTest questionnaire. Table 3 shows an overview of the data.

The questionnaire feedback suggests that young people may not have seen a significant benefit from undertaking the QbTest although not all had received feedback on the test at the time they completed the QbTest questionnaire. Data from the qualitative interviews also suggested that while some young people had been fine at the beginning of the test, they had found the process long and at times boring or felt exhausted by the end. One young person had to repeat the test and admitted feeling cross by this although another said he would recommend QbTest to other young people. At the time of the interview, not all young people had received the outcome of the test and ADHD assessment. One young person reported feeling excited by the possibility of a diagnosis and happy that he would be receiving help. Another young person was told that he did not have ADHD after the QbTest assessment but was unhappy with this as he did not feel this was correct.

Feedback from staff was generally more positive although they identified some challenges in using QbTest within a YOI setting. Initially, there had been some concerns because of the equipment (laptop and camera) and IT system needed but it was felt that it was easy to address. The main difficulties appeared to be finding an appropriate room for the equipment which was quiet for the testing environment. While it had been planned that all QbTests would be undertaken in the same room and the equipment set up; staff did have to move to other rooms on occasion. One member of staff said that it was easy to carry the equipment around and set up in another location if needed. There were problems with the prison staff facilitating the appointments in healthcare (eg, challenges in bringing the young people over to healthcare for their appointment). The healthcare staff often had to offer multiple appointments for the young person to attend to take the QbTest.

The staff members interviewed were of the opinion that QbTest was helpful in assessing ADHD and one staff member also reported that it helped both the young person and staff understand the young person's behaviours better. Staff felt that the QbTest could be helpful in improving waiting times and worked well alongside the ADHD clinical assessment. One staff member felt the QbTest was more objective and there was less bias than self-report measures. All the staff interviewed felt that the QbTest should be used in other sites as there were many advantages of using the QbTest.

**Table 2** Descriptive statistics for secondary outcome measures

| | Baseline (n=60) | | | | 3 months (n=47) | | | | 6 months (n=19) | | | |
|---|---|---|---|---|---|---|---|---|---|---|---|---|
| | QbTest | | Usual care | | QbTest | | Usual care | | QbTest | | Usual care | |
| | N | % | n | % | n | % | n | % | n | % | n | % |
| **SDQ total cut-off** | | | | | | | | | | | | |
| Close to average | 7 | 23.3 | 5 | 16.7 | 2 | 6.7 | 4 | 13.3 | 2 | 6.7 | 3 | 10.0 |
| Slightly raised | 4 | 13.3 | 8 | 26.7 | 0 | 0 | 5 | 16.7 | 0 | 0 | 4 | 13.3 |
| High | 2 | 6.7 | 5 | 16.7 | 4 | 13.3 | 1 | 3.3 | 0 | 0 | 1 | 3.3 |
| Very high | 16 | 53.3 | 12 | 40.0 | 7 | 23.3 | 7 | 23.3 | 7 | 23.3 | 1 | 3.3 |
| Missing | 1 | 3.3 | 0 | 0 | 17 | 56.7 | 13 | 43.3 | 21 | 70.0 | 21 | 70.0 |
| **B-BAARS** | | | | | | | | | | | | |
| Likelihood of receiving a clinical diagnosis (DSM-V criteria)‡§¶**†† | | | | | | | | | | | | |
| Yes | 21 | 70.0 | 23 | 76.7 | 10 | 33.3 | 12 | 40.0 | 7 | 23.3 | 4 | 13.3 |
| No | 8 | 26.7 | 7 | 23.3 | 3 | 10.0 | 6 | 20.0 | 2 | 6.7 | 4 | 13.3 |
| Missing | 1 | 3.3 | 0 | 0 | 17 | 56.7 | 12 | 40.0 | 21 | 70.0 | 22 | 73.3 |
| **SNAP*** | | | | | | | | | | | | |
| Missing | 26 | 86.7 | 27 | 90 | 30 | 100 | 30 | 100 | 30 | 100 | 30 | 100 |
| **CHU-9D*** | | | | | | | | | | | | |
| Missing | 2 | 6.7 | 0 | 0 | 17 | 56.7 | 13 | 43.3 | 21 | 70.0 | 22 | 73.3 |
| **C-GAS†** | | | | | | | | | | | | |
| Some noticeable problems | 3 | 10.0 | 0 | 0 | 0 | 0 | 0 | 0 | 0 | 0 | 0 | 0 |
| Some problems | 3 | 10.0 | 0 | 0 | 1 | 3.3 | 0 | 0 | 0 | 0 | 0 | 0 |
| Doing all right | 1 | 3.3 | 1 | 3.3 | 0 | 0 | 2 | 6.7 | 0 | 0 | 1 | 3.3 |
| Doing well | 0 | 7 | 2 | 6.7 | 0 | 0 | 2 | 6.7 | 0 | 0 | 0 | 0 |
| Missing | 23 | 76.7 | 27 | 90.0 | 29 | 96.7 | 26 | 86.7 | 30 | 100 | 29 | 96.7 |
| **Service use (CSRI)** | | | | | | | | | | | | |
| **Emergency attendance physical health** | | | | | | | | | | | | |
| Yes | 4 | 13.3 | 8 | 26.7 | 2 | 6.7 | 1 | 3.3 | 2 | 6.7 | 1 | 3.3 |
| No | 23 | 76.7 | 22 | 73.3 | 11 | 36.7 | 16 | 53.3 | 7 | 23.3 | 7 | 23.3 |
| Missing | 3 | 10.0 | 0 | 0 | 17 | 56.7 | 13 | 43.3 | 21 | 70.0 | 22.0 | 73.3 |
| **Planned admission physical health** | | | | | | | | | | | | |
| Yes | 1 | 3.3 | 0 | 0 | 0 | 0 | 0 | 0 | 0 | 0 | 0 | 0 |
| No | 26 | 86.7 | 30 | 100 | 13 | 43.3 | 17 | 56.7 | 9 | 30.0 | 8 | 26.7 |
| Missing | 3 | 10.0 | 0 | 0 | 17 | 56.7 | 13 | 43.3 | 21 | 70.0 | 22 | 73.3 |

Continued

**Table 2** Continued

| | Baseline (n=60) | | | | 3 months (n=47) | | | | 6 months (n=19) | | | |
|---|---|---|---|---|---|---|---|---|---|---|---|---|
| | QbTest | | Usual care | | QbTest | | Usual care | | QbTest | | Usual care | |
| | N | % | n | % | n | % | n | % | n | % | n | % |
| **Outpatient appointments physical health** | | | | | | | | | | | | |
| Yes | 2 | 6.7 | 2 | 6.7 | 1 | 3.3 | 4 | 13.3 | 0 | 0 | 1 | 3.3 |
| No | 25 | 83.3 | 28 | 93.3 | 12 | 40.0 | 13 | 43.3 | 9 | 30.0 | 7 | 23.3 |
| Missing | 3 | 10.0 | 0 | 0 | 17 | 56.7 | 13 | 43.3 | 21 | 70.0 | 22 | 73.3 |
| **Missed appointments** | | | | | | | | | | | | |
| Yes | 6 | 20.0 | 5 | 16.7 | 3 | 10.0 | 1 | 3.3 | 1 | 3.3 | 0 | 0 |
| No | 21 | 70.0 | 25 | 83.3 | 10 | 33.3 | 16 | 53.3 | 8 | 26.7 | 8 | 26.7 |
| Missing | 3 | 10.0 | 0 | 0 | 17 | 56.7 | 13 | 43.3 | 21 | 70 | 22 | 73.3 |
| **Emergency attendance mental health** | | | | | | | | | | | | |
| Yes | 1 | 3.3 | 2 | 6.7 | 0 | 0 | 0 | 0 | 0 | 0 | 0 | 0 |
| No | 26 | 86.7 | 28 | 93.3 | 13 | 43.3 | 17 | 56.7 | 9 | 30.0 | 8 | 26.7 |
| Missing | 3 | 10.0 | 0 | 0 | 17 | 56.7 | 13 | 43.3 | 21 | 70.0 | 22 | 73.3 |
| **Planned admission mental health** | | | | | | | | | | | | |
| Yes | 0 | 0 | 0 | 0 | 0 | 0 | 0 | 0 | 0 | 0 | 0 | 0 |
| No | 24 | 80.0 | 30 | 100.0 | 13 | 43.3 | 16 | 53.3 | 9 | 30.0 | 8 | 26.7 |
| Missing | 6 | 20.0 | 0 | 0 | 17 | 56.7 | 14 | 46.7 | 21 | 70.0 | 22 | 73.3 |
| **Missed appointments mental health** | | | | | | | | | | | | |
| Yes | 1 | 3.3 | 1 | 3.3 | 1 | 3.3 | 2 | 6.7 | 0 | 0 | 0 | 0 |
| No | 25 | 83.3 | 29 | 96.7 | 12 | 40.0 | 15 | 50 | 9 | 30.0 | 7 | 23.3 |
| Missing | 4 | 13.3 | 0 | 0 | 17 | 56.7 | 13 | 43.3 | 21 | 70.0 | 23 | 76.7 |
| **Number of service contacts (CSRI)** | | | | | | | | | | | | |
| **Physical health contacts** | | | | | | | | | | | | |
| Median | 3 | | 2 | | 4 | | 1 | | 2 | | 1 | |
| IQR | 1–8 | | 1–4 | | 1–8 | | 0–6 | | 2–3 | | 0–12 | |
| Min–Max | 0, 47 | | 0, 73 | | 0, 88 | | 0, 14 | | 0, 6 | | 0, 51 | |
| n | 27 | | 30 | | 13 | | 17 | | 9 | | 8 | |
| **Mental health contacts** | | | | | | | | | | | | |
| Median | 6 | | 4 | | 2 | | 5 | | 2 | | 0.5 | |
| IQR | 2–20 | | 1–6 | | 2–4 | | 4–12 | | 1–5 | | 0–1.5 | |
| Min–Max | 0, 89 | | 0, 333 | | 0, 18 | | 0, 24 | | 0, 37 | | 0, 5 | |

Continued

**Table 2** Continued

| | Baseline (n=60) | | | | 3 months (n=47) | | | | 6 months (n=19) | | | |
|---|---|---|---|---|---|---|---|---|---|---|---|---|
| | QbTest | | Usual care | | QbTest | | Usual care | | QbTest | | Usual care | |
| | N | % | n | % | n | % | n | % | n | % | n | % |
| n | 27 | | 30 | | 13 | | 17 | | 9 | | 8 | |
| **Criminal justice contacts** | | | | | | | | | | | | |
| Median | 9 | | 8.5 | | 4 | | 4 | | 2 | | 5 | |
| IQR | 3–23 | | 5–20 | | 3–6 | | 3–6 | | 1–4 | | 2.5–26 | |
| Min–Max | 1, 107 | | 1, 106 | | 0, 65 | | 0, 16 | | 0, 137 | | 2, 60 | |
| n | 27 | | 30 | | 13 | | 17 | | 9 | | 8 | |
| **Number of incidents** | | | | | | | | | | | | |
| Median | 1 | | 1 | | 2 | | 2 | | 3.5 | | 7 | |
| IQR | 0–2 | | 0–2 | | 1–5 | | 0–6.5 | | 1–4 | | 2–7 | |
| Min–Max | 0, 12 | | 0, 19 | | 0, 12 | | 0, 19 | | 1, 7 | | 0, 8 | |
| n | 29 | | 29 | | 9 | | 12 | | 6 | | 5 | |

*Missing data presented only.
†C-GAS has 10 categories from extremely impaired to doing very well-relevant categories presented only.
‡Five or more symptoms of inattention and/or ≥5 symptoms of hyperactivity/impulsivity must have persisted for ≥6 months to a degree that is inconsistent with the developmental level and negatively impacts social and academic/occupational activities.
§Several symptoms (inattentive or hyperactive/impulsive) were present before the age of 12 years.
¶Several symptoms (inattentive or hyperactive/impulsive) must be present in ≥2 settings (eg, at home, school, or work; with friends or relatives; in other activities)
**There is clear evidence that the symptoms interfere with or reduce the quality of social, academic, or occupational functioning
††Symptoms do not occur exclusively during the course of schizophrenia or another psychotic disorder, and are not better explained by another mental disorder (eg, mood disorder, anxiety disorder, dissociative disorder, personality disorder, substance intoxication, or withdrawal)
B-BAARS, Brief Barkley Adult ADHD Rating Scale; C-GAS, Children's Global Assessment Scale; CHU-9D, Child Health Utility Instrument; CSRI, Client Service Receipt Inventory; DSM-V, Diagnostic Statistical Manual of Mental Disorders, 5th Edition; SDQ, Strengths and Difficulties Questionnaire; SNAP, Swanson, Nolan and Pelham Rating Scale.

**Table 3** Descriptive statistics for the QbTest feedback questionnaire

| Item | Disagree | Neither | Agree |
|---|---|---|---|
| The QbTest results helped me understand my symptoms | 2 | 6 | 2 |
| The QbTest results were difficult to understand | 1 | 7 | 2 |
| I fully understood the purpose of the QbTest | 1 | 0 | 9 |
| I found the assessment very stressful | 1 | 0 | 9 |
| The task took too long to complete | 0 | 1 | 9 |
| I found the task difficult to complete | 1 | 1 | 8 |
| Overall the experience was useful | 3 | 5 | 2 |
| I found the chair very uncomfortable | 3 | 5 | 2 |
| The QbTest helped me understand the changes in my symptoms since my previous appointment* | 1 | 7 | 1 |
| I found the QbTest much easier to complete when on meds† | 0 | 6 | 0 |
| When the clinician talked through the output with me, it helped me understand how s/he had reached their diagnosis* | 0 | 6 | 3 |
| When the clinician talked me through the output, it helped me understand how they had reached their decision about medication | 0 | 6 | 2 |

*Missing data for one young person.
†Missing data for four young people.

One staff member added that the QbTest was helpful in the assessment of complex young people, where there might be concerns about co-morbid diagnosis.

## DISCUSSION, LIMITATIONS AND LESSONS LEARNT
### Findings
Despite the impact of COVID-19 restrictions, this study has provided important information regarding the feasibility of introducing QbTest into the secure estate and conducting a larger trial. Our primary outcomes were largely achieved; estimates of eligibility were broadly accurate, consent rates were high, we reached our recruitment target on time, randomisation and trial participation was high. Not all participants randomised to QbTest, undertook the test; primarily due to the impact of COVID-19 restrictions or staffing resources. We were able to follow-up young people, but this was easier over 3 months than 6 months and where the young people stayed within the YOI. Our secondary outcomes were more mixed; completion of outcome measures by young people was good overall at 3 months and 6 months where the young person remained in the secure estate or was contactable, but much poorer across all time points for measures completed by clinical staff and teachers. Only 23% of young people received a decision on diagnosis while they were within the secure estate which were all exclusion of ADHD. This is significantly lower than the number of diagnostic decisions made at 6 months from a community-based study (76% with QbTest plus usual ADHD care pathway and 50% with the usual ADHD care pathway alone).[22] This may reflect both the complexity of the needs of young people in the secure estate and the challenges in undertaking assessments within this environment.

Participants (both young people and staff) were largely supportive of the study and the role of QbTest. We know, from clinical practice, that young people can find QbTest challenging as it tests areas that they struggle with. This was evident from the questionnaire feedback and the interviews. However, this has also been reported in other studies and is experienced in clinical practice[23] and did not prevent young completing the test once they had started.

### Strengths and weaknesses
The study has contributed to developing the evidence base in understanding barriers and facilitators to implementing QbTest within a secure estate setting and has produced important feasibility information on eligibility, recruitment, attrition rates and collection of primary and secondary outcomes.

However, there were a number of limitations. The sample was obtained from only one site within the secure estate and includes boys only. Similar to the Assessing QbTest Utility in ADHD (AQUA) trial,[22] we allowed the clinical team to complete QbTest at a time that suited them. Although we had suggested that it would be ideal for the clinical team to undertake the QbTest within 2 weeks from randomisation the average number of days between randomisation and QbTest was much longer. Also, from the appointment data it suggested that not all young people were receiving usual care within the timeframes described by the clinical team at the start of the study and many young people did not receive any appointments. Young people not receiving appointments

were across both trial arms and throughout recruitment, although it was exacerbated by the impact of COVID-19. Staffing shortages, appointments not being attended, short sentences and the rigidity of the prison regime were also contributing factors to fewer diagnostic decisions being made. Recently, a consensus group of UK experts highlighted the issues within current ADHD care pathways,[35] given the complexity of the CYPSE it is likely that these issues are exacerbated.

## CONCLUSION

Aspects of this feasibility trial demonstrated that a larger RCT trial of the effectiveness of QbTest within the CYPSE could be feasible, with time to decision (days) as the primary outcome measure and secondary measures completed by the young person. However, features of the intervention implementation were mixed, partly exacerbated by the impact of COVID-19. The study highlighted challenges with the delivery of the usual ADHD care pathway within the secure estate which has implications for clinicians and commissioners. Prior to progressing to a trial, additional programme developmental work would need to be undertaken with staff working in the secure estate. This would involve the review of the usual care ADHD pathway to understand barriers to timely delivery and the development of a standardised ADHD care pathway across the CYPSE as well as strategies to improve collection of some secondary outcome measures. This would aim to increase the number of appointments young people have with healthcare teams and subsequently decisions on ADHD status, enabling a better estimate of a time to decision for usual care time and the required sample size for a large multisite RCT.

**Author affiliations**
[1]Children and Young People's Research Unit, Pennine Care NHS Foundation Trust, Ashton-under-Lyne, UK
[2]Psychiatry, Institute of Mental Health, NIHR CLAHRC-East Midlands, Nottingham, UK
[3]Centre for Biostatistics, Division of Population Health, Health Services Research and Primary Care, The University of Manchester, Manchester, UK
[4]Medical School, The University of Manchester, Manchester, UK
[5]Social Policy Research Unit, University of York, York, UK
[6]Department of Clinical and Forensic Psychology, Psychology ServicesLimited, London, UK
[7]Department of Psychology, University of Reykjavik, Reykjavik, Iceland
[8]Children and Adolescent Mental Health, Rotherham Doncaster and South Humber Mental Health NHS Foundation Trust, Doncaster, UK
[9]Child and Adolescent Mental Health Service, Pennine Care NHS Foundation Trust, Ashton-under-Lyne, UK
[10]Research andDevelopment, Leeds Community Healthcare NHS Trust, Leeds, UK
[11]Manchester Academic Health Science Centre, University of Manchester, Manchester, UK

**Acknowledgements** The active involvement of all young people and staff who participated within this research study and provided advice is gratefully acknowledged.

**Contributors** PC: Chief investigator, conceptualisation, supervision, methodology, writing and editing. CLH: Conceptualisation, methodology, writing—review and editing. L-AC: Statistician, conceptualisation, methodology, analysis, writing—review and editing. BB: Conceptualisation, methodology, writing—review and editing. SY: Conceptualisation, methodology, writing—review and editing. AK: Conceptualisation, methodology, writing—review and editing. ST: Patient and public involvement representative, analysis, writing—review and editing. LW-C: Researcher, data collection, writing—review and editing. MR: Researcher, data collection, data entry, analysis, writing—review and editing. VM: Researcher, data collection, data entry, analysis, writing—review and editing. CL: Project manager, conceptualisation, methodology, data entry, analysis, writing and editing.

**Funding** The study is funded by the NIHR RfPB (Grant number: PB-PG-1216-20007).

**Disclaimer** The views expressed are those of the author(s) and not necessarily those of the NHS, the NIHR of the Department of Health. The study sponsor and funders have no role in study design, including collection, management, analysis and interpretation of data; writing of the report and the decision to submit the report for publication

**Competing interests** None declared.

**Patient and public involvement** Patients and/or the public were involved in the design, or conduct, or reporting, or dissemination plans of this research. Refer to the Methods section for further details.

**Patient consent for publication** Not applicable.

**Ethics approval** This study involves human participants. All participants (or their parent or legal guardian) gave written informed consent to participate in the study. Ethical approval for the study was obtained from the Health and Research Care Wales; REC reference 18/WA/0347 and IRAS project ID 238947. Participants gave informed consent to participate in the study before taking part.

**Provenance and peer review** Not commissioned; externally peer reviewed.

**Data availability statement** No additional data is available due to the population of research participants involved in this study.

**ORCID iDs**
Prathiba Chitsabesan http://orcid.org/0000-0002-0968-3472
Charlotte Lucy Hall http://orcid.org/0000-0002-5412-6165
Charlotte Lennox http://orcid.org/0000-0001-9014-9965

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
