## [Reviewer comments · BMJ Open]

ARTICLE DETAILS

TITLE (PROVISIONAL)	Using an objective computer task (QbTest) to aid the identification of Attention Deficit Hyperactivity Disorder (ADHD) in the Children and Young People Secure Estate (CYPSE): A feasibility randomised controlled trial
AUTHORS	Chitsabesan, Prathiba; Hall, Charlotte; Carter, Lesley-Anne; Reeves, Mindy; Mohammed, Vaseem; Beresford, Bryony; Young, Susan; Kraam, Abdullah; Trowse, Sally; Wilkinson-Cunningham, Lloyd; Lennox, Charlotte

VERSION 1 – REVIEW

REVIEWER	Pia Tallberg Lund University, Clinical Science Lund
REVIEW RETURNED	16-Jun-2022

GENERAL COMMENTS	I know have reviewed your manuscript for the second time. I acknowledge the improvement of the article concerning the issues raised. I think the article is important and that you point out important improvements need to be done in the CYPSE. However, there are still some issues about the literature that need some amendment. 1. The chosen litterature to underpin the diagnostic utility of Qbtest still lack articles showing concerns using Qbtests, as well as other CPTs, diagnostically. Note that I do not argue about the Qbtest or the articles that do show clinical utility. It is the nuance that needs to be clarified. One concern is that some people with ADHD do not show impairment in psychometric tests, Qbtest or other CPTs. Another concern is that the Qbtest has shown difficulties identifying ADHD in clinical samples. Here are some references showing concerns about Qbtests: Adamou M, Jones SL, Marks L, Lowe D. Efficacy of Continuous Performance Testing in Adult ADHD in a Clinical Sample Using QbTest. J Atten Disord. 2022 Mar 7;10870547221079798. doi: 10.1177/10870547221079798. Epub ahead of print. PMID: 35255743. Baader A, Kiani B, Brunkhorst-Kanaan N, Kittel-Schneider S, Reif A, Grimm O. A Within-Sample Comparison of Two Innovative Neuropsychological Tests for Assessing ADHD. Brain Sci. 2020 Dec 31;11(1):36. doi: 10.3390/brainsci11010036. PMID: 33396421; PMCID: PMC7824145. Brunkhorst-Kanaan N, Verdenhalven M, Kittel-Schneider S, Vainieri I, Reif A, Grimm O. The Quantified Behavioral Test-A Confirmatory Test in the Diagnostic Process of Adult ADHD? Front Psychiatry. 2020 Mar 20;11:216. doi: 10.3389/fpsy.2020.00216. PMID: 32265761; PMCID: PMC7100366. Reh V, Schmidt M, Lam L, Schimmelmann BG, Hebebrand J, Rief W, et al. Behavioral Assessment of Core ADHD Symptoms Using the QbTest. Journal of attention disorders. 2015;19(12):1034-45.
---

	Tallberg P, Råstam M, Wenhov L, Eliasson G, Gustafsson P. Incremental clinical utility of continuous performance tests in childhood ADHD - an evidence-based assessment approach. Scand J Psychol. 2019 Feb;60(1):26-35. doi: 10.1111/sjop.12499. Epub 2018 Nov 19. PMID: 30452083; PMCID: PMC7379623. (1-3). Some references that raise concerns of using any CPT diagnostically:  1. Riccio CA, Reynolds CR, Lowe P, Moore JJ. The continuous performance test: a window on the neural substrates for attention? Archives of Clinical Neuropsychology. 2002;17(3):235-72. 2. Pineda DA, Puerta IC, Aguirre DC, Garcia-Barrera MA, Kamphaus RW. The role of neuropsychologic tests in the diagnosis of attention deficit hyperactivity disorder. Pediatric Neurology. 2007;36(6):373-81. 3. Preston AS, Fennell EB, Bussing R. Utility of a CPT in diagnosing ADHD among a representative sample of high-risk children: a cautionary study. Child neuropsychology : a journal on normal and abnormal development in childhood and adolescence. 2005;11(5):459-69. I think there are ways to use Qbtest (as well as other CPTs) in the diagnostic process. This is a challenge since there are some evidence that performance-based tests and rating scales of ADHD behaviour measuring different aspects of the disorder. And the diagnosis ADHD is a behavioral disorder. One proposed way of increasing the clinical utility in the diagnostic process is to use a step-wise fashion: Jarrett MA, Van Meter A, Youngstrom EA, Hilton DC, Ollendick TH. Evidence-Based Assessment of ADHD in Youth Using a Receiver Operating Characteristic Approach. Journal of clinical child and adolescent psychology : the official journal for the Society of Clinical Child and Adolescent Psychology, American Psychological Association, Division 53. 2016:1-13. Tallberg P, Råstam M, Wenhov L, Eliasson G, Gustafsson P. Incremental clinical utility of continuous performance tests in childhood ADHD - an evidence-based assessment approach. Scand J Psychol. 2019 Feb;60(1):26-35. doi: 10.1111/sjop.12499. Epub 2018 Nov 19. PMID: 30452083; PMCID: PMC7379623. (1-3).
--	--

VERSION 1 – AUTHOR RESPONSE

Reviewer: 1

Dr. Pia Tallberg, Lund University

Comments to the Author:

3. I know have reviewed your manuscript for the second time. I acknowledge the improvement of the article concerning the issues raised. I think the article is important and that you point out important improvements need to be done in the CYPSE. However, there are still some issues about the literature that need some amendment.

1. The chosen literature to underpin the diagnostic utility of Qbtest still lack articles showing concerns using Qbtests, as well as other CPTs, diagnostically. Note that I do not argue about the Qbtest or the articles that do show clinical utility. It is the nuance that needs to be clarified. One concern is that some people with ADHD do not show impairment in psychometric tests, Qbtest or other CPTs. Another concern is that the Qbtest has shown difficulties identifying ADHD in clinical samples. Here are some references showing concerns about Qbtests: Adamou M, Jones SL, Marks L, Lowe D.

Efficacy of Continuous Performance Testing in Adult ADHD in a Clinical Sample Using QbTest. *J Atten Disord.* 2022 Mar 7;10870547221079798. doi: 10.1177/10870547221079798. Epub ahead of print. PMID: 35255743.

Baader A, Kiani B, Brunkhorst-Kanaan N, Kittel-Schneider S, Reif A, Grimm O. A Within-Sample Comparison of Two Innovative Neuropsychological Tests for Assessing ADHD. *Brain Sci.* 2020 Dec 31;11(1):36. doi: 10.3390/brainsci11010036. PMID: 33396421; PMCID: PMC7824145.

Brunkhorst-Kanaan N, Verdenhalven M, Kittel-Schneider S, Vainieri I, Reif A, Grimm O. The Quantified Behavioral Test-A Confirmatory Test in the Diagnostic Process of Adult ADHD? *Front Psychiatry.* 2020 Mar 20;11:216. doi: 10.3389/fpsy.2020.00216. PMID: 32265761; PMCID: PMC7100366.

Reh V, Schmidt M, Lam L, Schimmelmann BG, Hebebrand J, Rief W, et al. Behavioral Assessment of Core ADHD Symptoms Using the QbTest. *Journal of attention disorders.* 2015;19(12):1034-45.

Tallberg P, Råstam M, Wenhov L, Eliasson G, Gustafsson P. Incremental clinical utility of continuous performance tests in childhood ADHD - an evidence-based assessment approach. *Scand J Psychol.* 2019 Feb;60(1):26-35. doi: 10.1111/sjop.12499. Epub 2018 Nov 19. PMID: 30452083; PMCID: PMC7379623.

(1-3).

Some references that raise concerns of using any CPT diagnostically:

1. Riccio CA, Reynolds CR, Lowe P, Moore JJ. The continuous performance test: a window on the neural substrates for attention? *Archives of Clinical Neuropsychology.* 2002;17(3):235-72.
2. Pineda DA, Puerta IC, Aguirre DC, Garcia-Barrera MA, Kamphaus RW. The role of neuropsychologic tests in the diagnosis of attention deficit hyperactivity disorder. *Pediatric Neurology.* 2007;36(6):373-81.
3. Preston AS, Fennell EB, Bussing R. Utility of a CPT in diagnosing ADHD among a representative sample of high-risk children: a cautionary study. *Child neuropsychology : a journal on normal and abnormal development in childhood and adolescence.* 2005;11(5):459-69.

• Response

Thankyou for highlighting studies that raise concerns about the use of CPT diagnostically and QbTest specifically.

We have now made changes to pages 4 and 5 of the document by 1) adding some of the references you have recommended above; 2) adding the concerns you have raised that not all people with ADHD show impairment on CPTs including QbTest and 3) difficulties that QbTest has shown in differentiating ADHD in clinical samples. However, we have also highlighted that many of the studies referenced use QbTest as a diagnostic tool to differentiate groups in the absence of additional clinical information. US FDA approval for QbTest is to augment but not replace standard clinical assessment. This emphasises the importance of using CPTs including QbTest alongside a clinical assessment (by an appropriately trained practitioner) which includes history, observation and informant information from multiple sources alongside any psychometric assessment.

I think there are ways to use Qbtest (as well as other CPTs) in the diagnostic process. This is a challenge since there are some evidence that performance-based tests and rating scales of ADHD behaviour measuring different aspects of the disorder. And the diagnosis ADHD is a behavioral disorder. One proposed way of increasing the clinical utility in the diagnostic process is to use a step-wise fashion: Jarrett MA, Van Meter A, Youngstrom EA, Hilton DC, Ollendick TH. Evidence-Based Assessment of ADHD in Youth Using a Receiver Operating Characteristic Approach. *Journal of clinical child and adolescent psychology : the official journal for the Society of Clinical Child and Adolescent Psychology, American Psychological Association, Division 53.* 2016:1-13. Tallberg P, Råstam M, Wenhov L, Eliasson G, Gustafsson P. Incremental clinical utility of continuous performance tests in childhood ADHD - an evidence-based assessment approach. *Scand J Psychol.* 2019 Feb;60(1):26-35. doi: 10.1111/sjop.12499. Epub 2018 Nov 19. PMID: 30452083; PMCID: PMC7379623.

(1-3).

• Response

We agree performance based tests and rating scales measure different aspects of ADHD and therefore see the potential benefit for including both in the assessment of ADHD in young people. Thankyou for highlighting these studies and the proposal of using a step wise approach to the diagnostic process by using QbTest at a later stage in the clinical care pathway when there is greater clinical uncertainty. We have referenced this approach on page 5 of the paper but explained why this is not recommended within the secure estate, as it may delay time to diagnostic decision and young people only remain within the secure estate for relatively short periods of time in comparison to young people in the community.

VERSION 2 – REVIEW

REVIEWER	Pia Tallberg Lund University, Clinical Science Lund
REVIEW RETURNED	17-Nov-2022
GENERAL COMMENTS	Dear author, I think the changes made improved the manuscript into a publishable fashion. I have no more comments.